# Dietary Administration of Black Raspberries and Arsenic Exposure: Changes in the Gut Microbiota and Its Functional Metabolites

**DOI:** 10.3390/metabo13020207

**Published:** 2023-01-30

**Authors:** Pengcheng Tu, Qiong Tang, Zhe Mo, Huixia Niu, Yang Hu, Lizhi Wu, Zhijian Chen, Xiaofeng Wang, Bei Gao

**Affiliations:** 1Department of Environmental Health, Zhejiang Provincial Center for Disease Control and Prevention, 3399 Binsheng Road, Hangzhou 310051, China; 2College of Standardization, China Jiliang University, Hangzhou 310018, China; 3School of Marine Sciences, Nanjing University of Information Science and Technology, Nanjing 210044, China; 4Key Laboratory of Hydrometeorological Disaster Mechanism and Warning of Ministry of Water Resources, Nanjing University of Information Science and Technology, Nanjing 210044, China

**Keywords:** black raspberries, arsenic, gut microbiota, gut microbiome, metabolites

## Abstract

Mounting evidence has linked berries to a variety of health benefits. We previously reported that administration of a diet rich in black raspberries (BRBs) impacted arsenic (As) biotransformation and reduced As-induced oxidative stress. To further characterize the role of the gut microbiota in BRB-mediated As toxicity, we utilized the dietary intervention of BRBs combined with a mouse model to demonstrate microbial changes by examining associated alterations in the gut microbiota, especially its functional metabolites. Results showed that BRB consumption changed As-induced gut microbial alterations through restoring and modifying the gut microbiome, including its composition, functions and metabolites. A number of functional metabolites in addition to bacterial genera were significantly altered, which may be linked to the effects of BRBs on arsenic exposure. Results of the present study suggest functional interactions between dietary administration of black raspberries and As exposure through the lens of the gut microbiota, and modulation of the gut microbiota and its functional metabolites could contribute to effects of administration of BRBs on As toxicity.

## 1. Introduction

Arsenic (As) exposure affects over 200 million members of the human population worldwide with geologically sourced contamination of drinking water being the major exposure route [1]. As exposure has been associated with a variety of human diseases including cardiovascular disease, diabetes, and bladder, lung, liver, and skin cancers [2]. It was previously established that perturbations of the gut microbiota and its metabolites could be a potential new mechanism by which As exposure leads to or exacerbates human diseases [3,4,5,6]. Therefore, modulation of the gut microbiota and its metabolic profile may affect As toxicity via impacting As-induced microbial perturbations.

Mounting evidence has indicated the essential role of gut microbiota in human health and disease [7]. The gut microbiome is involved in immune cell development, energy production, and epithelial homeostasis [8,9,10,11,12]. More importantly, metabolic activities of gut bacteria have been linked to xenobiotic metabolism and toxicity of environmental chemicals. For example, previous studies have indicated that gut bacteria are involved in metabolism and biotransformation of As [13], polycyclic aromatic hydrocarbons [14], and polychlorinated biphenyls [15]. Also, changes in the gut microbiota have been mechanistically associated with toxic effects of environmental agents such as heavy metals including As, artificial sweeteners, and pesticides [16]. We previously reported that administration of a BRB-rich diet substantially changes the mouse gut microbiome at both compositional and functional levels [17,18,19], suggesting the potential of black raspberries to modulate the gut microbiome. In addition, we previously reported that the BRB-rich diet impacted As biotransformation and reduced levels of oxidative stress in As-treated mice [20]. As shown in Appendix A, dietary administration of BRBs increased As methylation thereby elevating urinary total As in As-treated mice. On the other hand, by measurement of levels of 8-oxo-2′-deoxyguanosine, one of the most commonly-used biomarkers of oxidative DNA damage [21], results showed that BRBs attenuated As-induced oxidative stress in mice. These previous results together supported the involvement of BRB consumption in As metabolism and toxicity. Given the key role of the gut microbiome in As toxicity coupled with the effects of BRBs on As biotransformation/toxicity, it is of significance to elucidate the changes in the gut microbiome upon interactions of As exposure and dietary administration of BRBs.

In this follow-up study, we combined 16s rRNA gene sequencing and mass spectrometry-based metabolomics to extensively probe the changes in the gut microbiome of mice upon As exposure and dietary administration of black raspberries. Two complementary omic approaches were employed to achieve a comprehensive understanding of microbial changes. The 16S rRNA gene sequencing technique was used to identify bacterial changes in terms of abundance, which has been used as a mainstay of sequence-based bacterial analysis for decades. The other technique, activity-based metabolomics, allows the identification of metabolites that are differently abundant between groups with statistical significance. An untargeted approach enables the comprehensive comparison of metabolomes under different conditions, which is critical in understanding drivers of physiological activities related to gut microbiome. The results revealed that administration of BRBs reprogrammed the As-type gut microbiome, including alterations of various bacterial genera and key metabolites between groups with statistical significance, which could contribute to effects of BRBs on As biotransformation/toxicity. This follow-up study further elucidated changes of the gut microbiome in mice with dietary administration of black raspberries and As exposure, providing a connection with respect to diet, environmental agents, and the gut microbiota.

## 2. Materials and Methods

### 2.1. Preparation of Diets

A BRB-rich diet was prepared as previously described [17,18,19,20]. Briefly, whole ripe BRBs (*Rubus occidentalis*) of the Jewel variety were picked mechanically, washed with water, and frozen at −20 °C on a single farm within 2 to 3 h of picking. The harvested berries were then shipped frozen to Van Drunen Farms in Momence, Illinois, where they were freeze-dried under anoxic conditions to protect the integrity of berry components. Next, seeds were removed by forcing the freeze-dried berries through a small sieve, and the dried pulp was ground into powder. The berry powder was shipped to Ohio State University, where it was stored at −20 °C until further use. For standardization purposes, each batch of powder underwent a quantitative chemical analysis of 26 randomly selected nutrients and nonnutrient components [22,23]. The levels of the 26 components remain relatively stable compared to the initial analyses for at least 2 years in powder stored at −20 °C [23]. The BRB powder was stored at −20 °C until being incorporated into custom purified American Institute of Nutrition (AIN)-76A animal diet (Dyets, Inc., Bethlehem, PA, USA) by 10% *w*/*w* concentration at the expense of corn starch. AIN-76A diet was used as the control diet. Both diets were stored at 4 °C until being fed to mice.

### 2.2. Workflow to Investigate Functional Alterations of the Gut Microbiome by Dietary Administration of BRBs upon As Exposure

As reported in our previous study [20] (Appendix A), dietary administration of BRBs successfully increased urinary excretion of As as well as modulated As biotransformation via facilitating As methylation. Moreover, BRB consumption reduced levels of oxidative stress in mice induced by As exposure. In this follow-up study, we aimed to further investigate functional alterations of the gut microbiome in mice upon As exposure and dietary administration of BRBs. The experimental design is shown in Figure 1A; briefly, 40 mice were randomly assigned into 4 groups: 76, 76+ As, BRB, BRB+ As. Of these, the 76 and 76+ As groups were fed AIN-76A diet (control diet), while BRB and BRB+ As groups were fed BRB diet. After 2 weeks of dietary administration, 76+ As and BRB+ As groups were switched to be exposed to As via drinking water (10 ppm). After another 4 weeks of As treatment, fecal samples were collected for taxonomic characterization and metabolite profiling. The experimental workflow combined high-through 16S rRNA gene sequencing and mass-spectrometry-based metabolomics for the examination of changes in the gut microbiome resulting from BRB-mediated As toxicity.

### 2.3. Animals

The animal protocol was approved by the University of Georgia Institutional Animal Care and Use Committee (protocol No. A2013 06-033-Y3-A3). A total of 40 specific-pathogen-free (SPF) C57BL/6 mice (~8 weeks old) were purchased from Jackson Laboratories. The mice were housed in the animal facility of the University of Georgia. After 1 week of acclimation, mice were randomly assigned to 4 groups (76, 76+ As, BRB, BRB+ As). Of these, 76 and 76+ As groups were fed AIN-76A diet (control diet), while BRB and BRB+ As groups were fed BRB diet. Environmental conditions of 22 °C temperature, 40–70% humidity, with a 12/12 h light/dark cycle were applied. 76+ As and BRB+ As groups were exposed to As via drinking water (10 ppm) after 2 weeks. Mice of 30 g have an average daily water intake of 2 mL [24]. After another 4 weeks of As treatment, fecal samples were collected individually, and stored at −80 °C for further experiments.

### 2.4. 16S rRNA Gene Sequencing

Experiments of 16S rRNA gene sequencing were conducted as previously described [17]. Briefly, microbial DNA was extracted from mouse fecal samples (20–25 mg) using PowerSoil DNA isolation kit as per manufacturer’s instructions. For 16S rRNA gene sequencing, DNA was amplified using 515F and 806R primers to target the V4 regions of 16S rRNA gene. The DNA was then amplified, followed by normalization, barcoding, and the DNA was pooled, and quantified by Qubit 2.0 Fluorometer to construct the sequencing library. The resultant DNA was then paired-end sequenced using an Illumina MiSeq platform (Illumina, 500 cycles v2 kit, San Diego, CA, USA) in the Georgia Genomics Facility of University of Georgia. Paired-reads were assembled by the software Geneious (Biomatters, Auckland, New Zealand), followed by initial quality filtering with error probability of 0.01. The operational taxonomic unit (OTU) picking and diversity analysis was performed with a threshold of 97% sequence similarity by the software of Quantitative Insights into Microbial Ecology (QIIME). A representative sequence from each OTU was selected for taxonomic assignment according to Greengenes database (version 13_5; http://greengenes.lbl.gov/; accessed on 1 May 2019). By default, QIIME uses uclust consensus taxonomy classifier to assign taxonomy.

### 2.5. Untargeted Metabolomic Analysis

Experiments of untargeted metabolomic analysis were conducted as previously described [18]. Briefly, 20 mg of fecal samples and 50 mg of glass beads (Sigma-Aldrich, St. Louis, MO, USA) were mixed with 400 μL of cooled methanol solution (methanol/water 1:1). The mix was homogenized, and then centrifuged for 10 min at 12,000 rpm. The supernatant (~300 μL) was collected, dried in a SpeedVac (Savant SC110A; Thermo Electron, Waltham, MA, USA), and then resuspended in 30 μL 98:2 water:acetonitrile for MS analysis injection. LC-MS analysis was performed on a quadrupole time-of-flight (Q-TOF) 6530 (Agilent Technologies, Santa Clara, CA, USA) with an electrospray ionization source interfaced with an Agilent 1290 Infinity II UPLC system. The Q-TOF was calibrated daily using the standard tuning solution from Agilent Technologies. Metabolic features were analyzed in the positive ion mode using a C18 T3 reverse-phased column (Waters Corporation, Milford, MA, USA). The typical mass accuracy of the Q-TOF was <10 ppm. XCMS online server was applied for peak picking, alignment, integration, as well as extraction of peak intensities. MS/MS data were generated on Q-TOF for further identification of differently abundant features. Software packages MS-DIAL (version 2.9) and MS-FINDER (version 2.4) were applied for identification of metabolic features based on MS/MS spectrum [25,26].

### 2.6. Statistical Analysis

Alpha rarefaction and principal coordinate analysis (PCoA) were performed to assess alpha and beta diversities in the gut microbial communities, respectively. Alpha rarefaction analysis was performed using indices of observed OTUs, PD whole tree, and Chao1. PCoA was performed based on the unweighted UniFrac distance metric. Permutational multivariate analysis of variance (PERMANOVA) was applied to assess the difference between different cultivars. Also, principal component analysis (PCA) and hierarchical clustering algorithm were used for visualization of metabolite profiles. Differences in gut microbial abundances were assessed by a nonparametric test via Metastats. Two-tailed Welch’s *t*-test was used to analyze metabolites that were differently abundant between groups. False discovery rate (FDR) was used to correct multiple comparisons.

## 3. Results

### 3.1. Gut Microbial Changes at Compositional Level

Figure 1B,C shows the identified gut bacteria at order and family levels assigned from 16S rRNA sequencing reads with each color representing an individual bacterial order or family, respectively (legend at Appendix A). As exposure induced changes of consistent trends in several bacterial families regardless of different diets. For instance, in both 76 and BRB diet groups, Bifidobacteriaceae and Bacteroidales_f_S24-7 increased upon arsenic exposure. However, dietary difference contributes more significantly to microbiota changes compared to As exposure. Notably, Verrucomicrobiaceae was less than 0.1% in 76 diet groups, while in BRB groups, the proportions of Verrucomicrobiaceae were 51.5% and 30.2% in BRB and BRB+ As groups, respectively. Moreover, As also induced contradicted changes in different diet groups. For example, in 76 diet groups, Clostridiaceae decreased upon As exposure with a 1.5-fold change; however, in BRB diet groups, Clostridiaceae increased four-fold. Taken together, although diet contributes more significantly to gut microbial changes at compositional level, As exposure-induced gut microbiota changes differed if mice were fed different diets. In addition, Table 1 shows the fold changes of significantly-altered bacteria in mice upon arsenic exposure with different diets: there were nine bacterial genera that were significantly altered, with three increased and six decreased genera. Notably, compared with 76+ As group, Akkermansia increased with a fold change of approximate 5000 in BRB+ As group, which is consistent with effects of BRB on the gut microbiota from our previous report [17].

### 3.2. Principal Coordinate Analysis (PCoA) and Alpha Rarefaction Analysis

To further assess the differences of the gut bacterial community, alpha rarefaction and PCoA analyses were performed on mouse fecal samples to assess alpha and beta diversities in the gut microbial communities, respectively. Figure 2A shows the 3D PCoA plot of gut microbial communities. PCoA analysis based on the UniFrac distance metric reflects beta diversity between groups. The four sample groups were separated majorly driven by diet as indicated by separation between groups on different diets (*p* < 0.05). As exposure also impacts the gut microbial communities according to PCoA analysis, 17.28%, 6.84%, and 4.15% variation were explained by principal component (PC) 1, PC2, and PC3, respectively. In addition, alpha rarefaction analysis using indices of observed OTUs, PD whole tree, and Chao1 was shown in Figure 2B–D, respectively. Of these indices, observed OTUs and Chao1 reflect species richness in the community, and PD whole tree is a diversity calculated based on phylogenetic tree. Although there is no statistically significant difference, alpha diversities of the gut microbial community fluctuate upon arsenic exposure and dietary administration of BRBs.

### 3.3. Comparative Analysis of Metabolite Profiles

Figure 3A shows the principal component analysis (PCA) plot. The PCA was calculated using the feature intensities from all samples with colors assigned based on sample groups. Consistent with the PCoA result of gut microbial communities, diet plays a more significant role in separating different cultivars. Moreover, Figure 3B,C shows the cloud plots, constructed by altered features with green bubbles representing up-regulated features and red bubbles representing down-regulated features (*p*-value is represented by how dark or light the color is; fold change is represented by the radius of each feature; retention time is represented by position on the *x*-axis; mass-to-charge ratio is represented by position on *y*-axis). Compared with 76 group, BRB group had 1180 significantly-regulated features (Figure 3B). BRB+ As group had 958 features that were significantly changed compared to 76+ As group (Figure 3C). Metabolite profiling of the gut microbiome in mice with dietary administration of BRBs was reported in one of our previous studies [18]. In addition, the hierarchical clustering heat map constructed using intensities of shared features showed consistent patterns within individual groups (Figure 4). Not only As exposure induced microbial alterations in mice, but also these metabolic perturbations induced by As treatment were partly reversed by BRB dietary administration, indicating that administration of BRBs modulates and potentially restores As-treated gut microbiome and functional metabolites.

### 3.4. Key Metabolites Associated with Dietary Administration of BRBs upon As Exposure

To explore the role of BRB consumption in the gut microbiota of mice exposed to As, differently abundant metabolites between 76+ As and BRB+ As groups were profiled and identified. Table 2 lists the identified fecal metabolites. Between BRB+ As and 76+ As groups, there were a total of 18 identified metabolites that were significantly altered, including vitamin derivatives, bile acids, indoles, polyunsaturated fatty acids, bilirubins, and so forth. Of these metabolites, many of them are bioactive molecules that are involved in a number of metabolic processes and cellular functions. For example, flavins and tocopherols are vitamin derivatives related to the metabolic activities of gut bacteria and intestinal homeostasis. Riboflavin could be produced by the gut microbiota [27], and levels of tocopherols are associated with gut barrier functions [28]. Likewise, bile acids are a class of key metabolites for gut bacteria and diverse signaling pathways [29]. Taken together, key metabolites in the gut microbiota that were associated with dietary administration of BRBs upon As exposure could contribute to effects of BRB consumption on arsenic biotransformation/toxicity via host-gut microbiota axis.

## 4. Discussion

With the increasingly recognized role of the gut microbiome in As toxicity, the knowledge of how gut microbiome alterations affect health outcomes in As exposure is critical to development of therapeutic approaches via modulation of the gut microbiome. This knowledge is of importance, because it may be applied in the future to develop gut microbiome-targeted therapeutic approaches via dietary intervention. We previously reported that BRB consumption effectively affects As biotransformation and possibly impact As toxicity [20]. The objective of this follow-up study was to determine the role of the gut microbiota in BRB-mediated As biotransformation/toxicity. Given the essential role of the gut microbiome in As toxicity coupled with the effects of BRBs on As biotransformation/toxicity, illustration of changes in the gut microbiota of mice upon interactions of As exposure and dietary administration of black raspberries is of significance and represents an important step toward understanding how diet affects environmental exposure through the lens of the gut microbiota.

The gut microbiota not only directly impact intestinal homeostasis locally through microbial metabolic products, but also trigger systemic effects on remote tissues/organs such as the liver, adipose, or brain by producing metabolites that can act as signaling molecules [30,31]. Moreover, the role of the gut microbiota in transformation and metabolism of xenobiotics including As has been well recognized [3], indicated by previous reports on interactions of the gut microbiome with environmental toxic chemials such as As [3,13], diazinon [32], polycylic aromatic hydrocarbons [14], and polychlorinated biphenyls [15]. In addition, to perturb the gut microbiome and its functional metabolites is suggested to be a new mechanism of As toxicity [3,4,5,6]. Therefore, metabolic changes, especially perturbations in gut microbiota-related metabolites, play an essential role in As exposure and toxicity. In the present study, differently abundant metabolites in the gut microbiome of mice with or without administration of BRBs upon As exposure were profiled and identified. Significantly-altered metabolite fingerprints in fecal samples of mice were observed. Specifically, 12a-Hydroxy-3-oxocholadienic acid, belonging to the class of monohydroxy bile acids, alcohols and derivatives, increased with a fold change of two in fecal samples of mice fed BRBs compared to mice on control diet, upon As exposure. Bile acids are cholesterol derivatives synthesized in liver, which would undergo extensive enterohepatic recycling as well as modification by some gut bacteria. It is established that bile acids not only participate in digestion and absorption, but also serve as signaling molecules impacting a number of pathways by acting on diverse nuclear receptors [33]. Likewise, alpha-linolenic acid belongs to the class of lineolic acids and derivatives, which increased with a fold change of 1.8 in fecal samples of mice from BRB+ As group compared to mice from 76+ As group. Lineolic acids are polyunsaturated fatty acids with many beneficial effects associated with intestinal immunity and the gut microbiota [34]. Also, 13′-Carboxy-alpha-tocopherol is a derivative of tocopherol, which increased with a fold change of 1.5 in fecal samples of mice upon As exposure if fed BRBs compared to control diet. Tocopherols are known to confer protective effects on oxidative stress and inflammation [35,36,37]. More importantly, tocopherols may exhibit anti-cancer effect via several different cellular and molecular mechanisms [38], which could counter increased risks of bladder cancer and skin cancer associated with As exposure as well as possibly accounting for anti-cancer effects of BRBs, although the evidence remains controversial [39]. Thus, metabolic changes in the gut microbiota could contribute to effects of BRBs on mice upon As exposure.

The human gut microbiome contributes to human health and disease in a significant way, including key functions involved in immune cell development, energy production, and epithelial homeostasis [8,9,10,11,12]. Moreover, its role in transformation and metabolism of xenobiotics has been well recognized [3,16]. The gut microbiota continues to be an attractive therapeutic target. Currently, our knowledge of targeted and predictable modulation of the gut microbiome is in its infancy [40]. It is of significance to provide a diet-based approach for gut microbiome modulation, to elucidate how the modulated microbiome differently reacts to environmental chemicals, and to understand the role of microbiome-derived metabolites in these interactions [16,40]. Diet emerges as an essential determinant of gut microbial structure and functions [41]. Dietary modulation of the gut microbiome received considerable attention due to the advantages of low toxicity profiles and high patient compliance. Dietary recommendations to tackle gut microbiota-associated diseases such as inflammatory bowel disease (IBD) are usually based on inconclusive or controversial evidence [42], resulting from the complexity and variability of IBD disease pathogenesis including disease phenotype, gut microbiome, host genetic susceptibility, and environmental factors [43]. Thus, a therapeutic approach to treat these diseases through gut microbiome modulation is still highly desirable. On the other hand, previous studies showed that As exposure perturbed the gut microbiome and its metabolic functions [3], which may contribute to its toxicity. Modulation of the gut microbiome, in particular the microbial metabolites, could potentially alter As toxicity. Using a standardized BRB-rich diet and a mouse model, we have previously reported that BRB consumption effectively modulated the gut microbiota, including its composition, functions and metabolites [17,18,19]. Moreover, biotransformation and toxic effects of As were altered in mice that were fed BRBs [20]. Results of this follow-up study further supported the role of BRBs in mediating As toxicity by elucidating gut microbiota changes upon the interactions of dietary administration of black raspberries and arsenic exposure. Taken together, the potential of BRBs in modulating the gut microbiome, and specifically in intervening in the toxicity of environmental chemicals including As warrants future studies.

Admittedly this study was based on observation of effects of dietary administration of BRBs on mice upon As exposure and the gut microbiota, with no specific mechanism clearly illustrated. Further studies are needed to clarify the mechanism of interrelationships among those factors including the gut microbiome, its functional metabolites and brought effects for As-related adverse impact. Nevertheless, we profiled and identified metabolic changes in the gut microbiota of mice, and identified key metabolites that could contribute to effects of BRBs on mice upon As exposure. Although further investigation on mechanisms needs to be pursued, the present study is of significance for depicting the involvement of the gut microbiota in BRB-mediated As toxicity.

## 5. Conclusions

In the present study, we further analyzed changes in the gut microbiota and its functional metabolites upon interactions of As exposure and BRB administration to potentially identify microbial alterations that were mechanistically associated with BRB-mediated As toxicity. 16S rRNA gene sequencing and metabolomic profiling techniques were used to probe alterations in the gut microbiota and its metabolic profiles. The results clearly show that BRB significantly changed As-type gut microbiota at both compositional and functional levels. Microbial alterations induced by As exposure were restored or modified. In addition, alterations in a variety of gut microbiota-related metabolic products, including vitamin derivatives, bile acids, indoles, polyunsaturated fatty acids, and bilirubins, may be associated with effects on As toxicity by BRBs. Taken together, these findings may provide insights regarding the connection among diet, environmental exposure, and the gut microbiota, as well as offer evidence for future development of approaches for gut microbiome modulation.

## Figures and Tables

**Figure 1 metabolites-13-00207-f001:**
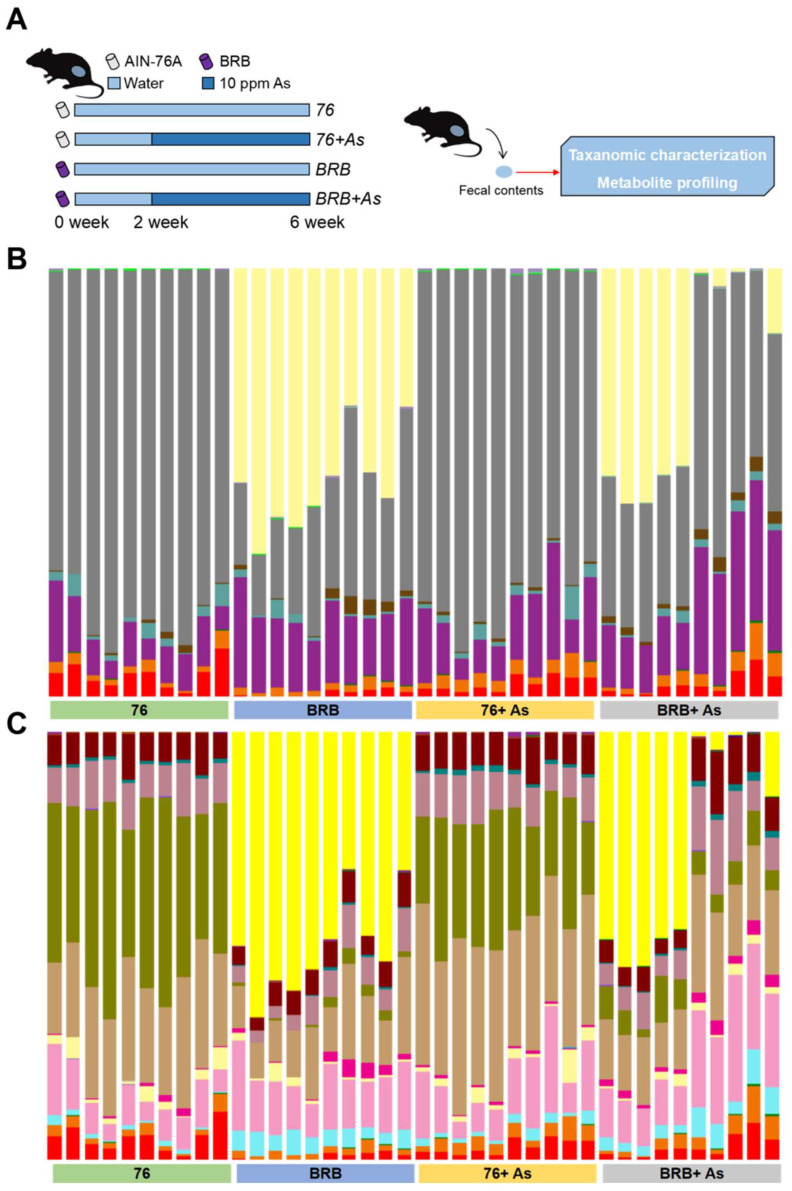
Taxonomic summaries of mouse gut microbial communities. (**A**) Experimental design. (**B**) Taxonomic summary at order level. (**C**) Taxonomic summary at family level (legend at Appendix A).

**Figure 2 metabolites-13-00207-f002:**
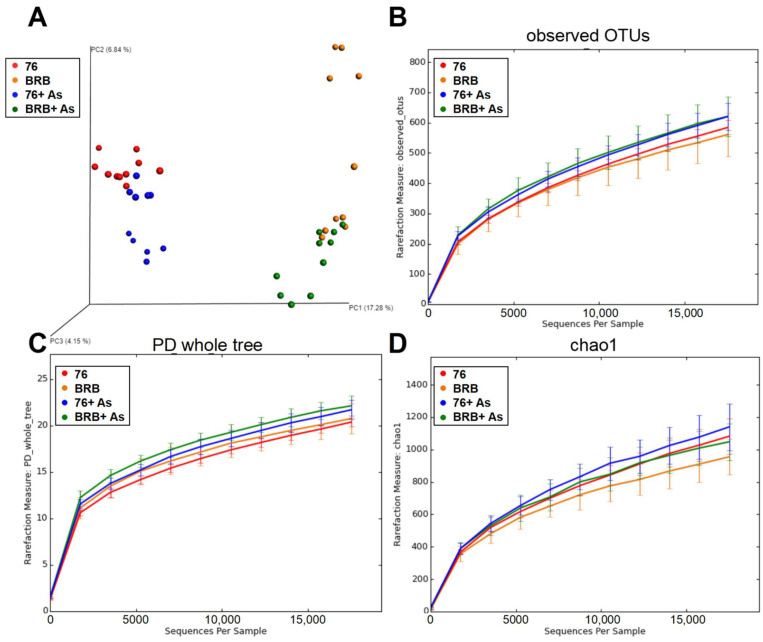
PCoA and alpha rarefaction analysis. (**A**) 3D PCoA plot of gut microbial communities, based on the unweighted UniFrac distance metric. Alpha rarefaction analysis using indices of observed OTUs (**B**), PD whole tree (**C**), and Chao1 (**D**).

**Figure 3 metabolites-13-00207-f003:**
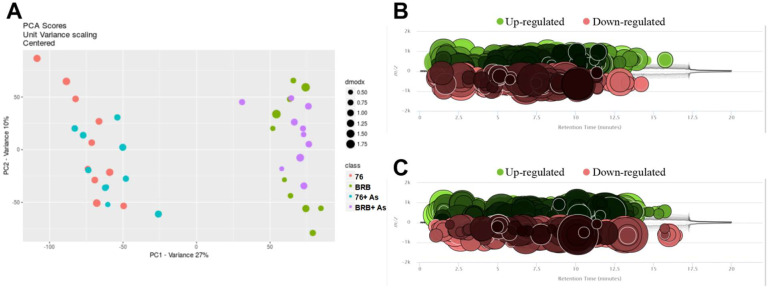
Comparative analysis of metabolite profiles. (**A**) PCA plot of four groups. Cloud plots constructed by altered features with green bubbles representing up-regulated features and red bubbles representing down-regulated features, 76 group versus BRB group (**B**), 76+ As group versus BRB+ As group (**C**).

**Figure 4 metabolites-13-00207-f004:**
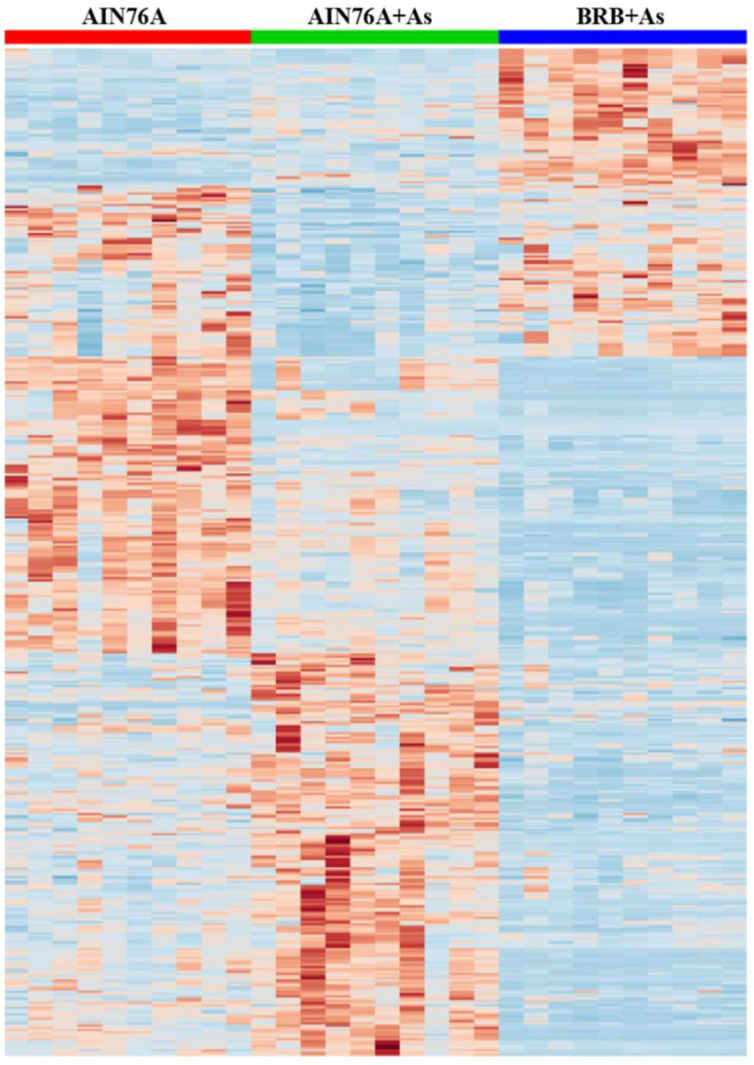
Hierarchical clustering heat map constructed using intensities of shared metabolite features (76, 76+ As, and BRB+ As groups) to display the pattern of modulation of BRBs on As-induced alterations in microbial metabolite fingerprints.

**Table 1 metabolites-13-00207-t001:** Effects of dietary administration of BRBs on gut microbial composition of mice upon As exposure.

Gut Bacteria	Mean Abundance (BRB+ As)	Mean Abundance (76+ As)	Up/Down	Fold Change	*p*-Value	*q*-Value
p_**Bacteroidetes**						
c_Bacteroidia; o_Bacteroidales; f_Rikenellaceae; g_	0.04249	0.01664	up	2.6	0.003	0.042
p_**Firmicutes**						
c_Bacilli; o_Turicibacterales; f_Turicibacteraceae; g_Turicibacter	0.01999	0.00503	up	4.0	0.001	0.025
c_Clostridia; o_Clostridiales; f_; g_	0.17616	0.32849	down	−1.9	0.001	0.025
c_Clostridia; o_Clostridiales; f_Clostridiaceae; g_	0.06063	0.25752	down	−4.2	0.001	0.025
c_Clostridia; o_Clostridiales; f_Clostridiaceae; Other	0.00006	0.00036	down	−6.0	0.001	0.025
c_Clostridia; o_Clostridiales; f_Lachnospiraceae; Other	0.00012	0.00073	down	−6.1	0.003	0.042
c_Clostridia; o_Clostridiales; f_Ruminococcaceae; g_Anaerotruncus	0.00004	0.00013	down	−3.3	0.003	0.042
c_Erysipelotrichi; o_Erysipelotrichales; f_Erysipelotrichaceae; g_Coprobacillus	0.00003	0.00019	down	−6.3	0.003	0.042
p_**Verrucomicrobia**						
c_Verrucomicrobiae; o_Verrucomicrobiales; f_Verrucomicrobiaceae; g_Akkermansia	0.27359	0.00005	up	5471.8	0.001	0.025

**Table 2 metabolites-13-00207-t002:** Effects of dietary administration of BRBs on microbial metabolites of mice upon As exposure. (n.a., the HMDB ID is not available for the corresponding metabolite).

Metabolites	Formula	*m*/*z*	Mean Intensity (76+ As)	Mean Intensity (BRB+ As)	Up/Down	Fold Change	Class	HMDB ID
Glutarylcarnitine	C_12_H_21_NO_6_	276.1507	1,889,595.8	4,226,664.3	up	2.2	Acyl carnitines	HMDB0013130
D-Urobilinogen	C_33_H_42_N_4_O_6_	591.3179	4,540,924.1	10,010,036.4	up	2.2	Bilirubins	HMDB0004158
D-Urobilin	C_33_H_40_N_4_O_6_	589.3026	7,015,796.9	18,248,399.2	up	2.6	Bilirubins	HMDB0004160
2-(3,4-dihydroxyphenyl)-8-[1-(2,4-dihydroxyphenyl)-3-(3,4-dihydroxyphenyl)-2-hydroxypropyl]-3,4-dihydro-2H-1-benzopyran-3,5,7-triol	C_30_H_28_O_11_	565.1689	383,161.4	27,032.6	down	−14.2	Catechins	n.a.
2,4-Toluenediamine	C_7_H_10_N_2_	123.0911	1,251,131.3	671,048.6	down	−1.9	Diaminotoluenes	HMDB0041799
Leucyl-phenylalanine	C_15_H_22_N_2_O_3_	279.171	1,583,370.0	9,250,466.2	up	5.8	Dipeptides	HMDB0302841
Trehalose 6-phosphate	C_12_H_23_O_14_P	423.0907	420,035.5	40,736.4	down	−10.3	Disaccharide phosphates	HMDB0001124
Riboflavin	C_17_H_20_N_4_O_6_	377.1487	1,382,839.9	831,508.2	down	−1.7	Flavins	n.a.
trans-Ferulic acid	C_10_H_10_O_4_	195.0674	47,598.4	22,572.8	down	−2.1	Hydroxycinnamic acids	n.a.
1H-Indole-3-carboxaldehyde	C_9_H_7_NO	146.0587	586,796.9	197,339.2	down	−3.0	Indoles	n.a.
Alpha-Linolenic acid	C_18_H_30_O_2_	279.2302	241,165.2	445,495.3	up	1.8	Lineolic acids and derivatives	HMDB0001388
(R)-lipoic acid	C_8_H_14_O_2_S_2_	207.0452	49,071.2	9526.7	down	−5.2	Lipoic acids and derivatives	n.a.
12a-Hydroxy-3-oxocholadienic acid	C_24_H_34_O_4_	387.2529	129,220.8	260,182.1	up	2.0	Monohydroxy bile acids, alcohols and derivatives	HMDB0000385
Coproporphyrin III	C_36_H_38_N_4_O_8_	655.2771	5,374,874.8	11,839,347.1	up	2.2	Porphyrins	HMDB0000570
Pyrrolidine	C_4_H_9_N	72.0799	350,852.3	1,307,818.2	up	3.7	Pyrrolidines	HMDB0031641
13′-Carboxy-alpha-tocopherol	C_29_H_48_O_4_	461.358	194,959.3	293,584.2	up	1.5	Tocopherols	HMDB0012555
Triethylamine	C_6_H_15_N	102.1287	636,910.8	1,085,599.8	up	1.7	Trialkylamines	HMDB0032539
4a-Carboxy-4b-methyl-5a-cholesta-8,24-dien-3b-ol	C_29_H_46_O_3_	443.35	396,953.2	655,442.3	up	1.7	Triterpenoids	HMDB0062383

## Data Availability

Not applicable.

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
