# Peer review of "Dietary Administration of Black Raspberries and Arsenic Exposure: Changes in the Gut Microbiota and Its Functional Metabolites"

_metabolites, 2023, doi:10.3390/metabo13020207_

Round 1
Reviewer 1 Report
The authors present a follow-up study about the effect of a Black Raspberries-rich diet on gut microbiome modulation after exposure to Arsenic.
The article is well-written and organized, and the results are interesting.
Figure 2 A is hard to read due to the light colours used. Please alter to more rich colours.
Figure 3 should include in the caption the colour code, although it is also explained in the text.
Figure 4 should also present the key metabolites from the mice fed with the BRB diet for comparison. What is the effect of just BRB on the gut microbiome and its faecal metabolites?
As toxicity includes, among others, an increased risk of bladder cancer and skin cancer. How are the protective effects of the BRB diet related to these diseases? Which metabolites could be associated with these types of cancers?
Reviewer 2 Report
the corrections are in file

Reviewer 3 Report
Tu et al. have previously reported that raspberries (BRBs) effectively affected arsenic (As) biotransformation and toxicity. In the present study, they have investigated the profiles of gut microbiome and metabolites in mice treated with arsenic (As) or combination of As and black raspberries (BRBs). Subsequently, they demonstrated that As treatment changes the profiles of gut microbiome and metabolites and BRBs treatment is able to change the As-induced alterations of gut microbiome and metabolites profiles. It might be interesting to focus gut microbiome and metabolites in this animal model. However, this study has just shown the investigated data. Thus, the relationships among gut microbiome, metabolites and brought effects for As-related harmful findings are not examined and discussed. The studies to clarify the mechanism of interrelationships among those factors at least would be needed. Without these trials, this study cannot give the researchers new findings or inspire them to hint the hypotheses.
Reviewer 4 Report
Tu et al.'s work was a multi-omics study to investigate the effect of BRBs on spf mice using 16S and LC-ms technologies to see BRB-mediated As toxicicity with microbiome and metabolome changes. They found observed microbiome-metabolome changes with BRB supplementation.
Comments:
Methods and results:
1. Microbiome analysis for PCOA, Fig 2, was performed which distancing methods? has there been any statistical analysis performed (e.g. PERMANOVA) ?? It is unclear to know from methods?
2. Similarly, with LC-MS, they i suggest the authors to follow HMDB to find InchI key identifiers to the metabolites instead of the iupac or chemical formula.
Minor comments:
1. Figure 1 legend is missing, which color represents which taxa? And same goes with the figure 4.
2. FIgure 2 can be made more readable? It is hard to follow.
Round 2
Reviewer 3 Report
In the revised version, authors have merely described their limitation shortly for the concerns from this reviewer. This never improve the previous version greatly.